# Detection of Gadolinium with an Impedimetric Platform Based on Gold Electrodes Functionalized by 2-Methylpyridine-Substituted Cyclam

**DOI:** 10.3390/s21051658

**Published:** 2021-02-28

**Authors:** Hassen Touzi, Yves Chevalier, Marie Martin, Hafedh Ben Ouada, Nicole Jaffrezic-Renault

**Affiliations:** 1Laboratory of Interfaces and Advanced Materials, Faculty of Sciences, University of Monastir, Monastir 5000, Tunisia; touzi_ha@yahoo.fr (H.T.); hafedhbenouada@gmail.com (H.B.O.); 2Laboratory for Process Control, Chemical Engineering and Pharmaceutical Engineering University of Lyon, LAGEPP UMR CNRS 5007, 43 Bd 11 Novembre 1918, 69622 Villeurbanne, France; yves.chevalier@univ-lyon1.fr; 3Institute of Analytical Sciences, University of Lyon, ISA UMR CNRS 5280, 5 Rue de La Doua, 69100 Villeurbanne, France; marie.martin@isa-lyon.fr

**Keywords:** gadolinium, impedance, chemical sensor, 2-methylpyridine-substituted cyclam

## Abstract

Gadolinium is extensively used in pharmaceuticals and is very toxic, so its sensitive detection is mandatory. This work presents the elaboration of a gadolinium chemical sensor based on 2-methylpyridine-substituted cyclam thin films, deposited on gold electrodes, using electrochemical impedance spectroscopy (EIS). The 2-methylpyridine-substituted cyclam (bis-N-MPyC) was synthesized in three steps, including the protection of cyclam by the formation of its CH_2_-bridged aminal derivative; the product was characterized by liquid ^1^H and ^13^C NMR spectroscopy. Spin-coated thin films of bis-N-MPyC on gold wafers were characterized by means of infrared spectroscopy in ATR (Attenuated Total Reflectance) mode, contact angle measurements and atomic force microscopy. The impedimetric chemical sensor was studied in the presence of increasing concentrations of lanthanides (Gd^3+^, Eu^3+^, Tb^3+^, Dy^3+^). Nyquist plots were fitted with an equivalent electrical circuit including two RC circuits in series corresponding to the bis-N-MPyC film and its interface with the electrolyte. The main parameter that varies with gadolinium concentration is the resistance of the film/electrolyte interface (*R_p_*), correlated to the rate of exchange between the proton and the lanthanide ion. Based on this parameter, the detection limit obtained is 35 pM. The bis-N-MPyC modified gold electrode was tested for the detection of gadolinium in spiked diluted negative urine control samples.

## 1. Introduction

Rare earth elements (REEs) occur in many parts of the Earth’s crust. The term “rare earths” is misleading because it refers to their low production and not to their abundance in the earth’s crust. In fact, the majority of REEs are more abundant than gold, silver or platinum [1]. The essential source of these REEs is open mineral deposits all over the world, especially in China and United States, although they can also be extracted from alternative sources such as industrial waste by recycling end-of-life consumer products [2]. REEs are used in “green” applications that are increasingly numerous due to their various chemical, electrical, magnetic, catalytic, metallurgical, optical and nuclear properties. They are found in the catalysts of the petroleum refining industry, in alloys for metallurgical processes, in glass for absorption of ultraviolet light and in electronics [3]. Other more futuristic applications of REEs include use in the production of renewable energies in high-capacity nickel–metal hydride batteries for hybrid cars [4], high-temperature superconductors, secure storage devices, transport of hydrogen for post-hydrocarbon economy [5], permanent magnets of industrial-scale wind turbines [6] and medical imaging as a contrast medium [7].

The extraction and treatment of REEs inevitably entails a major risk to the terrestrial ecosystem, the intensity of which depends on industrial processes as well as waste management. As a result, these processes lead to water pollution, destruction and poisoning of fauna and flora located around mineral deposits and industrial waste warehouses [8]. Despite this, these REEs are naturally present in soils, plants, waters and the atmosphere at low concentrations around the world, and they can accumulate in these environments in the form of anthropogenic inputs with low mobility [9]. For the last decade, the hazards of REEs have attracted the attention of the scientific community because of REE bioaccumulation in biota [10] and chronic toxicity [11].

Atomic absorption spectroscopy (AAS) [12], X-ray fluorescence spectrometry (XRF) [13], inductively coupled plasma mass spectroscopy (ICP-MS) [14], neutron activation analysis (NAA) [15] and inductively coupled plasma-optical emission spectrometry (ICP-OES) [16] are the main techniques reported for the sensitive detection of REEs. However, these laboratory methods require expensive equipment, multi-step sample preparation and trained professionals. Electrochemical sensors based on low cost and portable instrumentation are good candidates for the detection of REEs in the field. For the development of selective and sensitive chemical sensors, the design of selective REE-interacting assembled units is required in order to ensure the recognition layers at the surface of the electrochemical devices. Oxygen, sulfur and nitrogen heteroatoms allow the complexation of REE cations. Potentiometric sensors for different REEs were developed based on thiourea derivatives for the detection of Dy(III) [17], multidentate Schiff bases for the detection of Nd(III) [18], 4′-carboxybenzo-18-crown-6 for the detection of Lu(III) [19], 4-(3-nitrophenyl)-2,6-di-2-thienylpyridine for the detection of Tm(III) [20], 4-hydroxypyrrolidine-2-carboxylic acid for the detection of Nd(III) [21] or 2,2′-dithiobis(4-methylthiazole) for the detection of Lu(III) [22]. In all these cases, the ionophore molecules were inserted in a matrix such as poly(vinylchloride) or in carbon paste [22], thus limiting the shelf lifetime of the sensor due to the leakage of the ionophore molecule. Voltammetric sensors for different REs were developed, based on the modification of a gold electrode with 2-pyridinol-1-oxide [23], the insertion of nano-sized Eu^3+^-imprinted polymer in carbon paste electrodes for the detection of Eu(III) [24] or the insertion of nano-sized cerium-imprinted polymer in carbon paste electrodes for the detection of Ce(III) [25].

Gadolinium in the form of Gd^3+^ is an effective water proton relaxation reagent, used as a magnetic resonance imaging (MRI) contrast agent, due to its high paramagnetism [26]. This technique is widely used in medicine to achieve high resolution, three-dimensional images of the body. Free Gd^3+^ ions are highly toxic; thus, gadolinium should be injected in the form of soluble stable complexes. Complexes of Gd^3+^ with tetraazamacrocycles such as DOTA (1,4,7,10-tetraazacyclododecane-1,4,7,10-tetraacetic acid) are very stable even in the presence of endogenous available ions, due to kinetic inertia [27]. The excretion half-life of Gd complexes is much faster than that of free Gd^3+^ (5 min versus 7 days). Gd(III) is retained in the liver and then very slowly eliminated via the urinary tract. Consequently, due to the high toxicity of Gd(III) and the extensive use of gadolinium-based pharmaceuticals, the determination of gadolinium in urine is of outstanding importance. Its concentration in healthy subjects is as low as 0.3 µg∙L^−1^ [28].

These tetraazamacrocycles may differ by the size of the macrocycle cavity, the nature and the number of functional groups grafted onto the macrocycle. Indeed, they can be selectively functionalized by one or more chemical groups allowing the modification and the control of their coordination properties. This additional characteristic considerably increases their field of application and makes it possible to consider their use in various fields. Substituted cyclams attached to silicone materials were used for the functionalization of ISFET for the selective detection of ferric ions [29]. In recent years, the bisaminal process of tetraazamacrocycles (cyclen and cyclam) has been considered as one of the most powerful synthetic tools for the protection of amine functions during the N-mono- [30] and the N,N′-dialkylation [31] of several macrocycles using glyoxal reagents. This ligand is obtained by Royal’s method; it is the most efficient reagent using a tricyclic intermediate, including two amine functions [32]. Several electrochemical sensors for the detection of gadolinium have previously been reported. Voltammetric detection of gadolinium was achieved by the impregnation of nano-sized Gd(III)-imprinted polymer in carbon paste electrodes [33]. Impedimetric detection of gadolinium was obtained by the modification of gold electrodes with terpyridine ligands [34]. In this work, 2-methylpyridine-substituted cyclam (bis-N-MPyC) as a tetraazamacrocycle was used as a recognition molecule for the detection of gadolinium. It was synthesized using the bisaminal process and then immobilized as a film on gold electrodes. When a gadolinium solution is equilibrated with the modified gold electrode, due to the gadolinium complexation, the impedance of the modified gold electrode/electrolyte interface is modified, leading to sensitive detection of gadolinium.

## 2. Materials and Methods

### 2.1. Reagents

Cyclam (1,4,8,11-tetraazacyclotetradecane) from CheMatech (Dijon, France) was stored in vacuum desiccators. The 2-(chloromethyl) pyridine hydrochloride (picolyl chloride chlorohydrate) was purchased from Sigma-Aldrich. Formaldehyde from Sigma-Aldrich (37% in water, density 1.08) contained 10–15% of methanol as stabilizer to prevent polymerization. The solvents such as diethyl ether (C_2_H_5_)_2_O, dichloromethane (CH_2_Cl_2_), acetonitrile (CH_3_CN) and chloroform (CHCl_3_) from Sigma-Aldrich were anhydrous with a purity higher than 99.5%. Sodium iodide (NaI), magnesium sulphate (MgSO_4_) and ammonium acetate (CH_3_CO_2_^−^NH_4_^+^) were also purchased from Sigma-Aldrich. Four lanthanides were purchased from Sigma-Aldrich: gadolinium(III) nitrate hexahydrate (Gd(NO_3_)_3_·6H_2_O), europium(III) nitrate pentahydrate (Eu(NO_3_)_3_·5H_2_O), terbium(III) nitrate pentahydrate (Tb(NO_3_)_3_·5H_2_O) and dysprosium(III) nitrate hydrate (Dy(NO_3_)_3_·xH_2_O). Negative urine control was provided by Sigma-Aldrich.

### 2.2. Methods

Liquid state ^1^H and ^13^C NMR spectroscopy were performed using a Bruker Advance 500 spectrometer in CDCl_3_ solvent (δ in ppm from TMS). Fourier transform infrared spectra in ATR (Attenuated Total Reflectance) mode were recorded with a Nicolet iS50 FT-IR spectrometer. The surface energy of the functionalized gold electrode surface was evaluated from contact angle measurements using a goniometer from GBX Scientific Instruments (Romans, France). The AFM measurements were recorded using an Agilent 5500 atomic force microscope (Agilent Technologies, Palo Alto, CA, USA).

The gold substrates (small squares of 1 × 1 cm^2^) were supplied by the French RENATECH network (LAAS, CNRS Toulouse). The gold layer thickness was about 300 nm and was obtained by evaporation of gold on an oxidized silicon substrate (SiO_2_ layer of 300 nm thickness) capped with a 30 nm titanium adhesion layer. The electrochemical impedance spectroscopy was performed in a double-walled electrochemical cell fabricated from Pyrex glass (volume capacity of 5 mL) fitted with a conventional three electrode system: a modified gold electrode as working electrode (0.19 cm^2^ active area defined by an O-ring seal), a platinum counter-electrode and a saturated calomel electrode (SCE) used as a reference electrode. Impedance spectroscopy experiments were carried out at room temperature by using an “Autolab PGSTAT 302 N” impedance analyzer driven by analysis system software (FRA2) adapted to impedance measurements.

### 2.3. Procedures

#### 2.3.1. Synthesis of Bis-N-(Dimethylpyridine)-Armed Cyclam

The macrocycle carrying two-armed methylpyridine at positions 1 and 8 was synthesized using Royal’s method, which is the most efficient for the formation of a tricyclic intermediate bearing two amine groups [32]. The synthesis of bis-N-(dimethylpyridine) cyclam (bis-N-MPyC) was then carried out in three steps: the protection of amine groups, the dialkylation of the protected cyclam and then the deprotection of the amine groups.

The protection of amine groups was achieved by rapid addition of 0.26 g (2.4 eq.) of formaldehyde to 0.26 g (1 eq.) of cyclam solubilized in ultra-pure water under strong stirring. The reaction mixture was kept for 3 h at 0 °C in an ice bath. The obtained precipitate was filtrated and washed three times with 20 mL of diethyl ether before drying under vacuum. The resulting product was recovered as a white powder with a yield of 96%.

For the dialkylation step, 0.65 g of picolyl chloride chlorohydrate was deprotonated in a basic solution at pH 12, obtained by addition of NaOH. The organic phase was recovered after three extractions with 10 mL of dichloromethane each time in a separating funnel. It was dried under magnesium sulphate and filtered on glass wool before evaporation of dichloromethane solvent. The obtained viscous red product was added to 5 mL of acetonitrile before being mixed with a solution of protected cyclam (0.2 g in 5 mL of acetonitrile). Next, 0.6 g of sodium iodide (NaI) was added to this mixture which was stirred for 7 days at room temperature. A white ammonium di-iodide salt formed; it was washed with acetonitrile before drying under vacuum.

The deprotection of the amine groups was carried out by basic hydrolysis using sodium hydroxide. A total of 0.18 g of the obtained di-salt was added to 3 mL of an aqueous solution of NaOH (4 M). The reaction mixture was left under stirring for 8 h at room temperature. The resulting product was recovered after extraction three times with 10 mL of chloroform and dried under magnesium sulphate before evaporation of the solvent. A beige powder (bis-N-MPyC) was obtained with a yield of 84% (Figure 1).

#### 2.3.2. Elaboration of the Bis-N-MPyC Thin Film on Gold Electrodes

The gold wafers used as transducer were rinsed with acetone for 10 min at room temperature using a Branson 5210 ultrasonic bath (Branson Ultrasonics Corporation, Danbury, CT, USA) before rinsing with ultrapure water and drying under nitrogen flow. These wafers were cleaned in freshly heated piranha solution (H_2_O_2_/H_2_SO_4_ mixture 3:7 *v*/*v*) for about 2 min in order to activate the surface [35], rinsed with ultrapure water and then dried under nitrogen flow.

The immobilization of sensitive film on the cleaned gold surface was performed by means of spin-coating. A drop (10 μL) of bis-N-MPyC solution in chloroform (10^−2^ M) was deposited on a rotating gold wafer at a speed of 2000 rpm for 20 s. The resulting thin film on gold wafers was dried at 60 °C for 1 h under vacuum before being soaked in 0.01 M acetate buffer solution overnight to condition and stabilize the film thickness before being used in the electrochemical cell.

#### 2.3.3. Fourier Transform Infrared Spectroscopy (FTIR) in ATR Mode

FTIR-ATR spectroscopy was carried out to characterize the deposited bis-N-MPyC film on the clean gold wafers. A Nicolet iS50 FT-IR spectrometer combined with the ATR iS50 accessory for 400 scans at a resolution of 8 cm^−1^ was used. A background spectrum of the clean gold wafer and the spectrum of the gold-bis-N-MPyC film were obtained under the same conditions.

#### 2.3.4. Contact Angle Measurements

The surface free energy of Bis-N-MPyC functionalized Au wafers was determined from contact angle measurements of drops of three different liquids (water, formamide and diiodomethane), deposited on bare and functionalized gold surfaces. This wettability technique is based on the capture of succession of images by a CCD (charged-coupled device) camera connected to a graphics card of a drop (4 μL) of liquid probe deposited by a graduated syringe. Measurements were repeated 20 times. These digital images were analyzed, and the average contact angle was deduced from the right and left angles extracted from pictures of the drops at equilibrium by the tangent method.

The total surface free energy γ^s^ of the clean or Au/bis-N-MPyC functionalized gold wafers and its dispersive γ^d^ and polar γ^p^ components were determined from the contact angles of the three liquid probes based on the Owens–Wendt method. This method was supplemented by the van Oss method, which decomposes the polar γ^p^ component into its acid γ^+^ and basic γ^−^ components that describe the donor and acceptor characteristics of the surface [36].

#### 2.3.5. Surface Characterization Using Atomic Force Microscopy (AFM)

The surface topography of the bis-N-MPyC films adsorbed on the clean gold surfaces was assessed with AFM. The measurements were performed with an Agilent 5500 atomic force microscope (Agilent Technologies, Palo Alto, CA, USA). Silicon tips with a nominal spring constant of 20 nm^−1^ were used in tapping mode at a frequency of ~300 kHz.

#### 2.3.6. Impedance Spectroscopy (EIS)

EIS measurements were carried out at room temperature (20 ± 3 °C) in an electrochemical cell connected to an impedance analyzer and controlled by computer. The modified gold Au/bis-N-MPyC electrode, platinum counter-electrode and saturated calomel electrode (SCE) were carefully inserted into the cell to prevent trapping air bubbles and ensure sealing. Four mL of aqueous 0.01 M ammonium acetate (CH_3_CO_2_^−^NH_4_^+^) solution at pH 6.8 was used as background electrolyte. The measurement of the impedance was based on the measurement of the response of the interface between the electrolyte solution and the bis-N-MPyC film to a sinusoidal excitation potential with low amplitude (10 mV) applied between the working electrode and the reference electrode. The impedance spectra were recorded at different D.C. applied potentials ranging from −0.9 to −0.3 V in the frequency domain 0.1 Hz to 100 kHz to set the optimum polarization for stable measurements of the electrochemical impedance. The potential polarization used for these measurements was optimized at −0.9 V. At the gold surface, there is competition between the adsorption of hydroxyl anions (OH^−^) and acetate anions (AcO^−^). At pH between 6.5 and 7.5, the main redox reaction is: Au(OAc)_n_ + ne^−^ ⇔ Au + nOAc^−^. Under nitrogen flow, the reduction peak was observed at −0.8 V/SCE and the oxidation peak at −0.3 V/SCE. The redox activity of acetate anions was already observed at the platinum surface [37]. Once all these parameters were fixed, several measurements were repeated until the last Nyquist diagrams were superimposed. The first lanthanide was added to the electrochemical cell immediately, and measurements were carried out after 30 min for equilibration, according to our previous work [29]. Nyquist diagrams for each lanthanide concentration were recorded. The equivalent circuit parameters for the electrolyte interface of the functionalized surface of Au/bis-N-MPyC were directly deduced from the experimental data by applying the software FRA2, after testing several models of equivalent circuit. The impedimetric responses of bis-N-MPyC modified electrodes to the lanthanide ions Gd^3+^, Eu^3+^, Tb^3+^ and Dy^3+^ were investigated.

## 3. Results and Discussion

### 3.1. Synthesis and Liquid ^1^H and ^13^C NMR Characterization

As the (1,4,8,11-tetraazacyclotetradecane) cyclam contains four amine groups with the same reactivity with respect to electrophilic attack, protection of these groups is necessary to carry out alkylation. This was performed via the reaction of 2.4 equivalents of formaldehyde with cyclam. The protected cyclam obtained as a white powder was readily isolated by simple filtration. Bis-N-dialkylation is favored on the non-adjacent amine groups of the protected cyclam, and the obtained compound presents a trans configuration. This step was carried out by dissolving the protected cyclam in acetonitrile, after which 2 equivalents of picolyl chloride were rapidly added. Because of its ionic character, the dialkylated salt is insoluble in acetonitrile and was readily recovered by simple filtration. The addition of 3 equivalents of NaI in the reaction mixture was intended to substitute iodine for the chlorine atom of picolyl chloride, which is used to promote nucleophilic substitution in picolyl and improves the reaction yield. The deprotection of amine groups was carried out by basic hydrolysis using an aqueous solution of NaOH (4 M) stirred for 8 h at room temperature. The final product bis-N-MPyC and the formaldehyde-protected cyclam were characterized by liquid ^1^H and ^13^C NMR spectroscopy.

The ^1^H and ^13^C NMR spectra of the formaldehyde-protected cyclam (CDCl_3_, δ in ppm from TMS as a reference) showed the following peaks:

^1^H NMR: 1.14–1.18 (m, CH_2_ β-N); 2.19–2.28 (m, CH_2_ β-N); 2.34–3.14 (m, CH_2_ α-N); 5.40–5.42 (dt, aminal CH_2_).

^13^C NMR: 20.3 (CH_2_ β-N); 49.4 (CH_2_ α-N); 53.7 (CH_2_ α-N); 68.96 (aminal CH_2_).

The ^1^H and ^13^C NMR spectra of bis-N-MPyC (CDCl_3_, δ in ppm from TMS as a reference) after reaction of N-dialkylation of formaldehyde-protected cyclam with picolyl chloride chlorohydrate and deprotection of amines of the di-quaternary ammonium salt showed the following peaks:

^1^H NMR: 1.72 (m, CH_2_ β-N); 2.46–2.6 (m, CH_2_ α-N); 3.0 (s, NH); 3.71 (s, N-CH_2_-pyridine); 6.69–8.39 (aromatic protons of pyridine).

^13^C NMR: 25.8 (m, CH_2_ β-N); 47.3 (CH_2_ α-N); 49.3 (CH_2_ α-N); 51.6 (CH_2_ α-N); 54.2 (CH_2_ α-N); 59.2 (N-CH_2_-pyridine), 121.6–158.4 (aromatic carbons of pyridine).

The ^1^H NMR spectra before and after dialkylation and after basic hydrolysis reactions were compared. The peak at 5.42 ppm corresponding to the aminal CH_2_ protons in the spectrum before bis-N-dialkylation and basic hydrolysis disappeared once the reaction was completed. Several peaks appeared in the spectrum of bis-N-MPyC after dialkylation and basic hydrolysis reaction. The first one at 3.0 ppm was assigned to protons of the secondary amine groups (-NH-) that appeared after deprotection of the tertiary amine groups of cyclam. The second peak at 3.71 ppm was assigned to the protons of CH_2_ groups adjacent to pyridine groups and tertiary amines (N-CH_2_-pyridine). The peaks at 6.69–8.39 ppm were attributed to the aromatic protons of pyridine.

The ^13^C NMR spectrum of the resulting product bis-N-MPyC displayed a peak at 59.2 ppm corresponding to the carbon of CH_2_ groups adjacent to pyridine groups and tertiary amines (N-CH_2_-pyridine) and several peaks at 121.6–158.4 ppm corresponding to the aromatic carbons of pyridine. In addition, the disappearance of the peak at 68.96 ppm corresponding to carbons of aminal CH_2_ in the formaldehyde-protected cyclam was noticed in the final spectrum of the bis-N-MPyC.

### 3.2. Elaboration of the Bis-N-MPyC Functionalized Gold Electrodes

The bis-N-MPyC compound in chloroform was deposited on the cleaned gold surface by means of a spin-coating process. After drying operations, the wafers modified with bis-N-MPyC film were characterized by means of infrared spectroscopy in ATR mode, contact angle measurements and atomic force microscopy and analyzed by electrochemical impedance spectroscopy.

Adsorption of the bis-N-MPyC molecules to the surface of the gold electrodes is ensured by interactions between the π-electrons in pyridine groups of bis-N-MPyC and the gold atoms on the wafer surface (Figure 2). The interactions of conjugated bands in aromatic compounds with several metals have been investigated in detail [38]. The adsorbed aromatic compounds have been detected on metal gold wafers by the technique of surface enhanced Raman spectroscopy (SERS) [39].

### 3.3. Surface Characterization of the Functionalized Cyclam Film

#### 3.3.1. FTIR-ATR Mode Spectroscopy

The FTIR-ATR mode spectrum of the bis-N-MPyC film (Figure 3) showed several bands related to both methylpyridine groups and cyclam cavity such as tertiary and secondary amines. In this spectrum, the bands at 727–869 and 1746 cm^−1^ were attributed to δ(=CH) and δ(CH sp2) deformation vibrations, respectively, of the pyridine ring. Moreover, the bands at 1379–1464, 1570 and those at 3004 and 3054 cm^−1^ were attributed to ν(C=C), ν(C=N) and ν(CH sp2) stretching vibrations, respectively, in the same ring. The bands at 1003 and 1047; 1118; 2854–2963 and at 3400 cm^−1^ were attributed to ν(C-C), ν(C-NSec: secondary amines), ν(CH sp3: -CH_2_- groups) and ν(NH: secondary amines) stretching vibrations, respectively, for the cyclam macrocycle. These peaks were attributed in our previous work [29]. The bands at 1264 and at 1596 and 1671 cm^−1^ were attributed to ν(C-NSec: tertiary amines) stretching and to δ(NH: secondary amines) deformation vibrations, respectively. The observation of characteristic vibrations of secondary amines and tertiary amines showed that the cyclam was functionalized by dialkylation of its amino groups.

#### 3.3.2. Surface Wettability

The hydrophilic or hydrophobic characteristics on the surfaces were assessed by measurements of the contact angles between deposited water drops and bare gold wafers (or modified gold wafers with bis-N-MPyC films). Indeed, a significant decrease in the contact angle of water drops from 55° for the bare gold surface to 7.8° for the surface of bis-N-MPyC film was observed. This decrease of contact angle revealed its hydrophilic character, which is explained by the presence of the polar NH groups on the cyclam macrocycle. The total surface free energy γ^s^ is equivalent to the sum of dispersive γ^d^ and polar γ^p^ components calculated from the measurements of the contact angles of the different liquid probes (water, formamide and diiodomethane) deposited on the functionalized surfaces, according to the Owens–Wendt method. Alternatively, the polar component was converted to acid–base interactions according to the van Oss method giving two new acid γ^+^ and basic γ^−^ components. The total surface free energy is given by the following formula [40]:$$\gamma^{s}=\gamma^{d}+\gamma^{p}=\gamma^{d}+2{\left(\gamma^{+}\gamma^{-}\right)}^{1/2}$$

All different values of contact angles and surface energies of bare gold wafer and bis-N-MPyC film are presented in Table 1. A slight increase in total surface free energy from the bare gold surface to the bis-N-MPyC film was observed. This was due to the increase of its dispersive γ^d^ and polar γ^p^ components. A large increase of the basic component of the surface free energy γ^−^ was observed when the bare gold wafers were modified by bis-N-MPyC film. This basic component calculated from the van Oss method expresses the strong basic character of the bis-N-MPyC film, which is due to the presence of the amine groups of cyclam (Figure 2).

#### 3.3.3. Atomic Force Microscopy

An AFM image measured after the deposition of the bis-N-MPyC film on the gold surface is presented in Figure 4. It displays islands of around 1 to 2 µm diameter and around 40 to 57 nm height. This is evidence of the adsorption of multilayers of bis-N-MPyC molecules. The formation of such multilayers should involve hydrophobic interactions and π-stacking interactions between the naphthyl residues on the one hand, and attractive interactions between the cyclam residues on the other hand, the first layers being adsorbed as presented in Figure 2.

#### 3.3.4. Determination of the Coverage of Modified Gold Wafers

The coverage (*θ*) of the gold surface by functionalized cyclam film was deduced from the variation of the real part of the impedance (*Zr*) with respect to the inverse of the square root of the sinusoidal excitation pulsation ($\omega^{-\frac{1}{2}}$). Figure 5 shows such data for the three electrodes: gold, Au/bis-N-MPyC and Au/bis-N-MPyC-Gd^3+^. The extrapolation of the linear zone at high frequencies corresponds to the resistive part of the impedance. The coverages of 90.9% and 93.7% for the Au/bis-N-MPyC and Au/bis-N-MPyC-Gd^3+^ electrodes were calculated from the following relationship:θ=1−[Zr(gold electrode)/Zr(functionalized gold electrode)]

These high coverages of the gold surface by bis-N-MPyC and bis-N-MPyC-Gd^3+^ thin films can be explained by an almost complete coating of the gold surface. Indeed, bis-N-MPyC thin films were less dense and more porous than the bis-N-MPyC-Gd^3+^ thin films. This proved the formation of a coordination complex between the Gd^3+^ ions and bis-N-MPyC molecules.

### 3.4. Electrochemical Characteristics of the Functionalized Cyclam Electrode

#### 3.4.1. Impedance Measurements and Equivalent Circuit Modeling

Figure 6 shows the impedance characteristics of cleaned and functionalized gold electrodes as Nyquist plots. The three Nyquist diagrams correspond to the bare gold electrode (a), Au/bis-N-MPyC electrode (b) and Au/bis-N-MPyC-Gd^3+^ (10^−5^ M) electrode (c) immersed in acetate background electrolyte.

Nyquist diagrams were fitted with several models of electrical circuits using the data processing software FRA2 of the Autolab instrument until a satisfactory fit to the experimental data was reached. The equivalent electrical circuit presented in Figure 7 fitted at best all data of the Nyquist diagram of the Au/bis-N-MPyC electrode (Figure 6b) with good accuracy (chi-square of the order of 10^−^^2^). The parameters of this equivalent electrical circuit for the electrolyte/bis-N-MPyC/Au structure are *R_s_*, *R_p_*, *R_f_*, *CPE_dl_* and *CPE_f_*, representing respectively the ohmic resistance of the bulk electrolyte; the polarization resistance of the electrolyte/film interface; the resistance of the film; the constant phase element as a non-ideal capacitor of the electrolyte/film interface and the constant phase element as a non-ideal capacitor of the film. The constant phase elements *CPE* account for the interfacial irregularities such as roughness and porosity. Their impedance is expressed by the following relationship [41]:ZCPE=1Q(jω)n
where *Q* is a constant, *j* is the imaginary number, *ω* = 2*πf* is the angular frequency and *n* is a correction exponent (0 < *n* < 1). The *CPE* corresponds to a more capacitive behavior when the value of *n* tends to 1.

The values of the different parameters obtained from fitting the Nyquist plots are presented in Table 2. The mean value of the resistance of the film (*R_f_*) before contact with lanthanides is 24.5 ± 4.1 kΩ·cm^−^^2^ (RSD = 20%). From [42], it is established that the reproducibility of the film thickness deposited by the spin-coating technique is 10%. The discrepancy between the RSD of experimental thickness and the RSD of evaluated resistance is due to the approximation in the resistive and capacitive properties of the film. The obtained value of *n_f_* is 0.86 ± 0.01, showing that the film is not purely capacitive, due to its roughness and its porosity.

The increase of the semicircle diameters is related to the evolution of the polarization resistance *R_p_* that shifted from 7.2 kΩ·cm^−^^2^ (Figure 6a) for the bare gold electrode/electrolyte interface to 19.62 kΩ·cm^−^^2^ (Figure 6b) for the bis-N-MPyC/electrolyte interface. This difference of polarization resistance is explained by proton exchange process on the bis-N-MPyC film. During interfacial equilibration with the background electrolyte (0.01 M ammonium acetate solution at pH 6.8), amino groups in the functionalized cyclam film get protonated. However, the diffusion of the acetate ions of the background electrolyte into the bis-N-MPyC film compensates the charge of the protonated amine groups. The increase of the polarization resistance *R_p_* of the bis-N-MPyC-Gd^3+^ film/electrolyte interface up to 27.15 kΩ·cm^−^^2^ (Figure 6c) in the presence of Gd^3+^ ions can be explained by the ion-exchange mechanism of the protons and the Gd^3+^ ions. Such exchange promotes the formation of a gadolinium complex because the release of protons decreases the positive charge of the film. The overall charge neutrality is ensured by adsorption of acetate ions. The evolution of the resistance R*_f_* of bis-N-MPyC film from 25.59 to 39.20 kΩ·cm^−^^2^ in the presence of Gd^3+^ ions (10^−5^ M) confirmed this result.

This mechanism of ion exchange and complex formation is illustrated in Figure 8. The stability of the gadolinium complex, the charge neutrality of which is ensured by adsorption of acetate ions, may be due to the fastest water exchange rate reported in the literature for a neutral complex with only one inner sphere water molecule [43]. We observed that for pH values lower than 2, there is no formation of the gadolinium complex (no change of the Nyquist diagram), whereas the formation becomes fast when the pH value reaches the value of 4. This observation agrees with the results obtained in the study of kinetic behaviors of lanthanides mixed with DOTA in acetic buffer solution [44]. Stability of the gadolinium complex is obtained for pH values between 4 and 8, as shown for Gd complexes with benzimidazole-DOTA [45]. The pH value of 6.8 was chosen in this stability range, and in this pH range (4–8) the sensor response is stable.

The coordination of Gd^3+^ ion with -N- is shown by the shift of the δ(NH: secondary amines) from 1668 to 1673 cm^−1^ (Figure 9). Moreover, the intensity of this band was decreased due to the decrease of the number of N-H bonds. In comparison, the ν(CH sp3: -CH2-group) of the cyclam macrocycle at 2928 cm^−1^ was not shifted and remained constant.

#### 3.4.2. Effect of the Lanthanide Concentration on the Impedance of the N-MNCyclen Functionalized Gold Electrodes

Nyquist diagrams resulting from electrochemical impedance spectroscopy were plotted at different concentrations of each lanthanide (Gd^3+^, Eu^3+^, Tb^3+^ and Dy^3+^) that binds to Au/bis-N-MPyC electrodes (Figure 10). They exhibited a semicircular shape in the high frequency zone, which considerably depended on the concentration of lanthanides. Indeed, the diameters of the semicircles increased as a function of the concentration of the lanthanides to be detected. All the parameters of the equivalent electrical circuit for the electrolyte/bis-N-MPyC/Au at different lanthanide concentrations (*R_f_*, *CPE_f_*, *n_f_*, *R_p_*, *CPE_dl_* and *n_dl_*) and for the electrolyte/bis-N-MPyC/Au without lanthanide are reported in Table 2. The variation of the Nyquist diagrams was studied for the concentrations ranging from 10^−^^10^ M to 10^−^^5^ M of lanthanides. The resistance of the ammonium acetate support electrolyte (0.01 M at pH 6.8) was equal to *R_s_* = 236 ± 37 Ω·cm^−^^2^ for all injections of the lanthanides. The parameters that drastically changed when the concentration of lanthanide increased in the electrochemical cell were the polarization resistance *R_p_* of the bis-N-MPyC-lanthanide film/electrolyte interface (Figure 11) and the resistance *R_f_* of bis-N-MPyC-lanthanide film. However, the presence of the only Gd^3+^ caused *R_p_* to increase. *R_f_* increased with the same slope ((Δ*R_f_*/*R_f_*_0_)/decade) × 100 = 10) for all the studied lanthanides, which showed that the complexes had the same stability constant for all the studied lanthanides. The value of *R_p_* is correlated to the rate of exchange between the proton and the lanthanide ions; this exchange rate varies more rapidly with the Gd^3+^ concentration, which should be correlated to the stability of the neutral complex with only one inner sphere water molecule [43].

### 3.5. Analytical Performance of the Impedimetric Sensor

The relative variation of polarization resistance *R_pr_* = (*R_p_* − *R_p_*_0_)/*R_p_*_0_) was plotted as a function of the co-logarithm of the concentration of different lanthanides Gd^3+^, Eu^3+^, Tb^3+^ and Dy^3+^ as a calibration of the impedimetric sensor in 0.01 M ammonium acetate background electrolyte at pH 6.8 (Figure 10). The parameters *R_p_*_0_ and *R_p_* are the polarization resistance the bis-N-MPyC film/electrolyte interface before and after contact with different concentrations of lanthanides. The slopes and domain of linearity determined from these curves given in Table 3 highlight the higher sensitivity of bis-N-MPyC film to Gd^3+^ ions, with ∆*R_p_*/*R_p_*_0_ of Gd^3+^ being more than 7.3 times larger than for Eu^3+^, Tb^3+^ and Dy^3+^. A detection limit of 35 pM (5.5 ng∙L^−1^) was reached. This detection limit is quite compatible with the specifications for the detection of gadolinium in urine in radiology practice (0.3 µg∙L^−1^). The detection limit of this sensor is much lower than that of the previously published electrochemical and optical sensors for gadolinium (Table 4). The detection limit was in the order of a few pM, the same range as that of the sophisticated analytical technique ICP-MS. The reproducibility of the measurements carried out on three bis-N-MPyC modified electrodes was characterized by a relative standard deviation of less than 5%. When the bis-N-MPyC modified electrodes were stored dry at a temperature of 4 °C and the measurements were performed every day after washing with an EDTA solution, the sensitivity for the detection of gadolinium remained unchanged (±5%) over 6 months.

Detection in spiked negative urine control samples, diluted in 0.01 M ammonium acetate solution, was performed. For spiked concentrations of 8.0 and 20.0 nM, the detected concentrations obtained from the calibration curve (Figure 10) were 7.7 ± 0.3 and 21.1 ± 0.9 nM, respectively.

## 4. Conclusions

2-Methylpyridine-substituted cyclam (bis-N-MPyC) synthesized according to the bisaminal procedure was deposited as a film on a gold electrode. After its chemical and morphological characterization, the bis-N-MPyC modified gold electrode was used as an impedimetric sensor for the detection of lanthanides. The polarization resistance of the bis-N-MPyC film/electrolyte interface varied as a function of the log of the concentration of gadolinium(III) leading to sensitivity for the detection of Gd^3+^ of over 7.3 times that of Eu^3+^, Tb^3+^ and Dy^3+^. A detection limit of 35 pM was obtained, which is very low compared to that of the previously published electrochemical and optical sensors for Gd^3+^. This detection limit is in the right range for the detection of gadolinium in urine in the framework of medical applications. The obtained impedimetric sensor was tested for the detection of gadolinium in spiked diluted negative urine control samples.

## Figures and Tables

**Figure 1 sensors-21-01658-f001:**
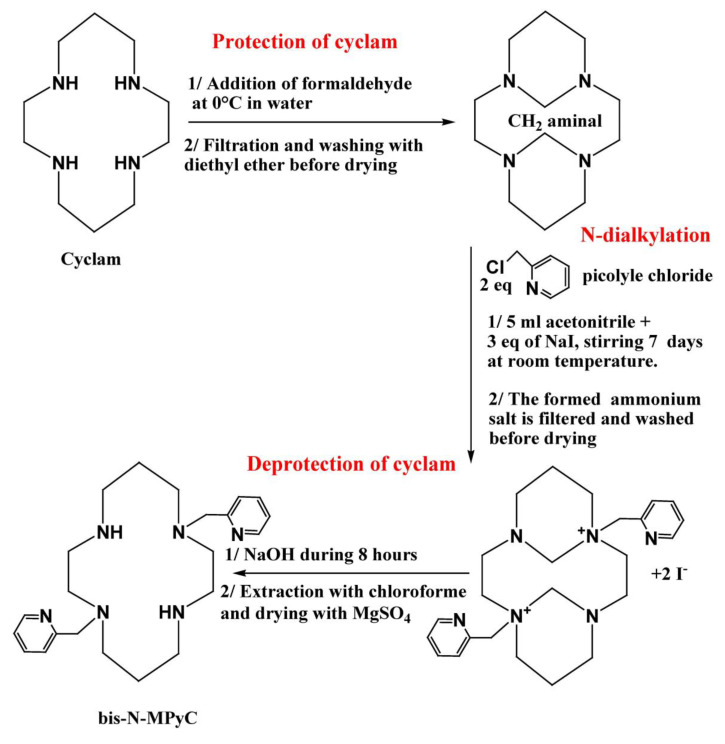
The synthesis of di-N-alkylated systems from cyclam.

**Figure 2 sensors-21-01658-f002:**
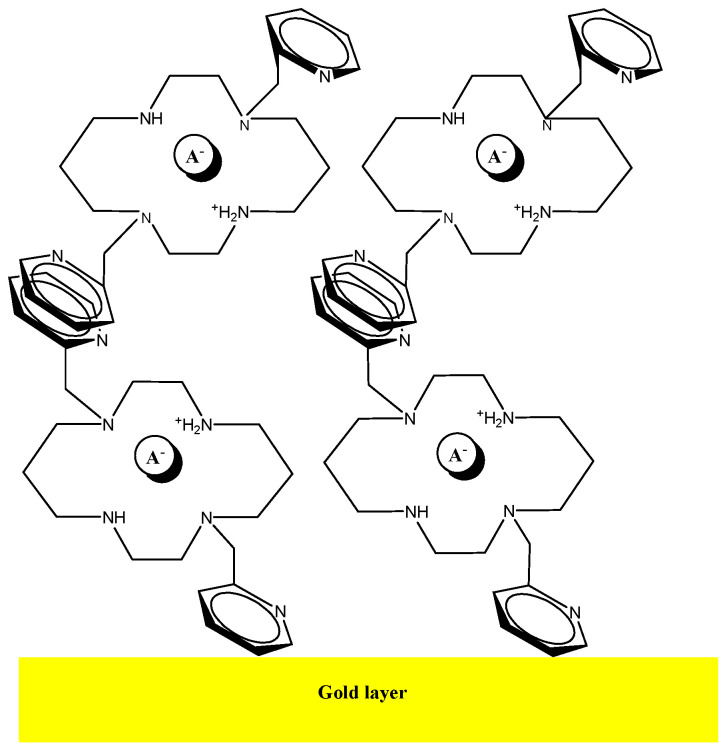
Schematic representation of the multilayer of bis-N-MPyC on the gold surface.

**Figure 3 sensors-21-01658-f003:**
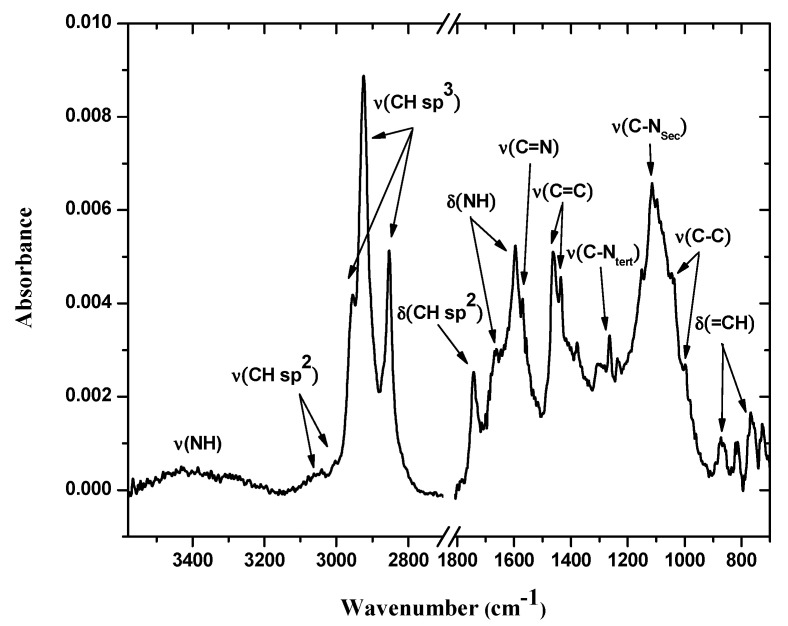
IR-ATR spectra of bis-N-MPyC film surface (from 700 to 1800 and from 2700 to 3600 cm^−^^1^).

**Figure 4 sensors-21-01658-f004:**
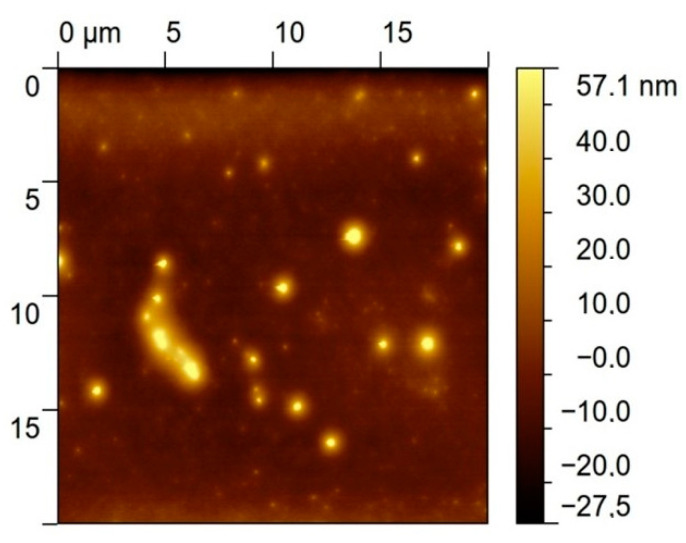
AFM image of the gold/bis-N-MPyC surface.

**Figure 5 sensors-21-01658-f005:**
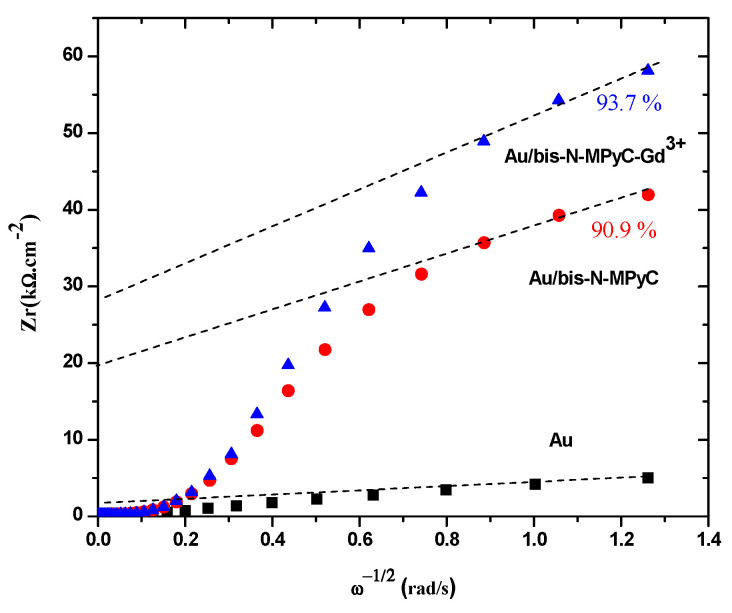
Determination of the coverage of bis-N-MPyC and bis-N-MPyC–Gd^3+^ thin films.

**Figure 6 sensors-21-01658-f006:**
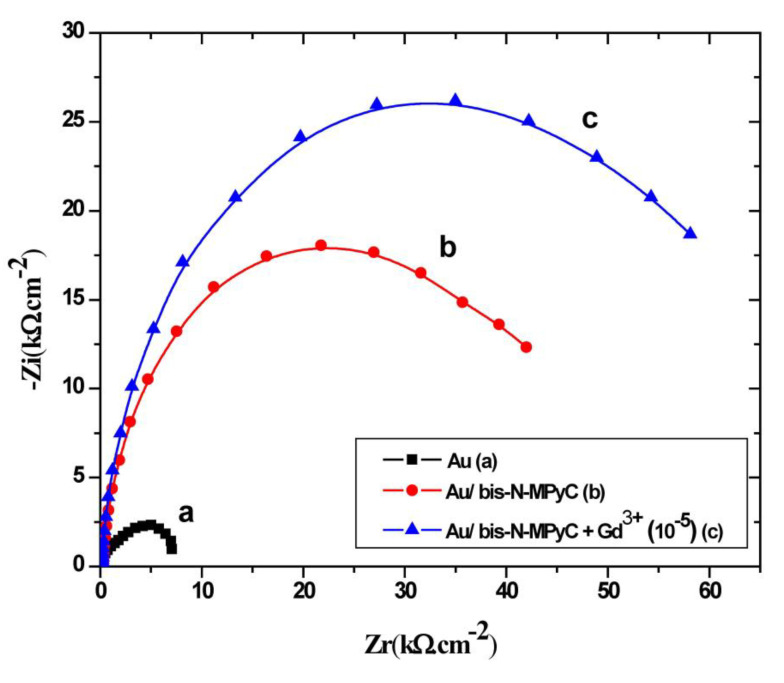
Nyquist diagrams at −0.9 V/SCE polarization, in the frequency domain 0.1 Hz to 100 kHz, for bare gold electrode (**a**), bis-N-MPyC-functionalized gold electrode (**b**) and Gd^3+^-bis-N-MPyC-functionalized gold electrode (**c**). Measurement medium: 0.01 M aqueous ammonium acetate (CH_3_CO_2_^−^ NH_4_^+^) solution at pH 6.8.

**Figure 7 sensors-21-01658-f007:**
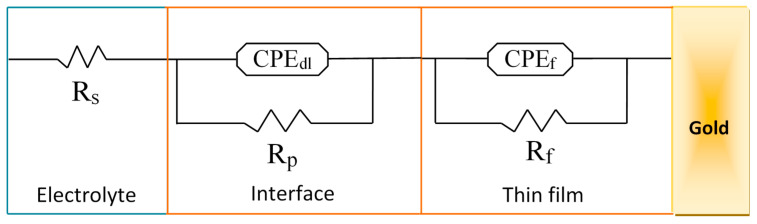
Equivalent circuit fitted to the impedance spectra, *R_s_*: resistance of the bulk electrolyte; *CPE_dl_*: constant phase element of the interface; *R_p_*: polarization resistance of the interface; *CPE_f_*: constant phase element of the film; *R_f_*: resistance of the film.

**Figure 8 sensors-21-01658-f008:**
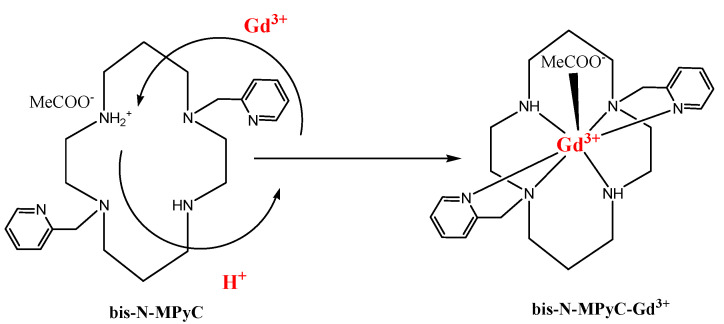
Proposed Gd^3+^ complexing process with functionalized cyclam ligands.

**Figure 9 sensors-21-01658-f009:**
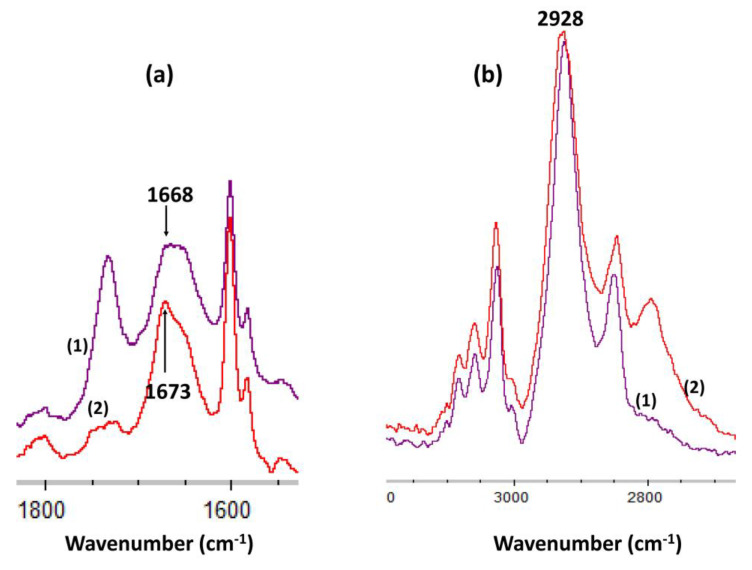
FTIR spectra of (**a**) δ(NH: secondary amines) band in bis-N-MPyC film (1) and in bis-N-MPyC film + Gd^3+^ (2) (**b**) ν(CH sp3: -CH2-group) band in bis-N-MPyC film (1) and in bis-N-MPyC film + Gd^3+^ (2).

**Figure 10 sensors-21-01658-f010:**
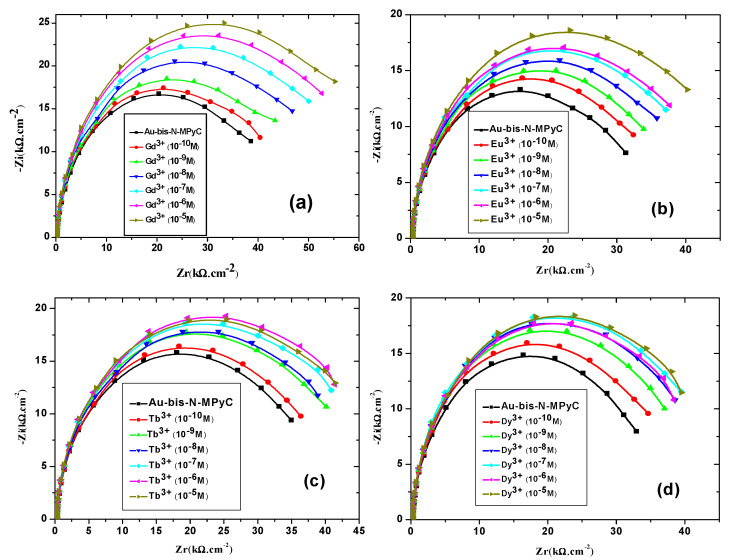
Nyquist diagrams of the gold electrodes coated with bis-N-MPyC thin film in ammonium acetate 10^−2^ M at pH 6.8 for different concentrations of lanthanides Gd^3+^ (**a**), Eu^3+^ (**b**), Tb^3+^ (**c**) and Dy^3+^ (**d**).

**Figure 11 sensors-21-01658-f011:**
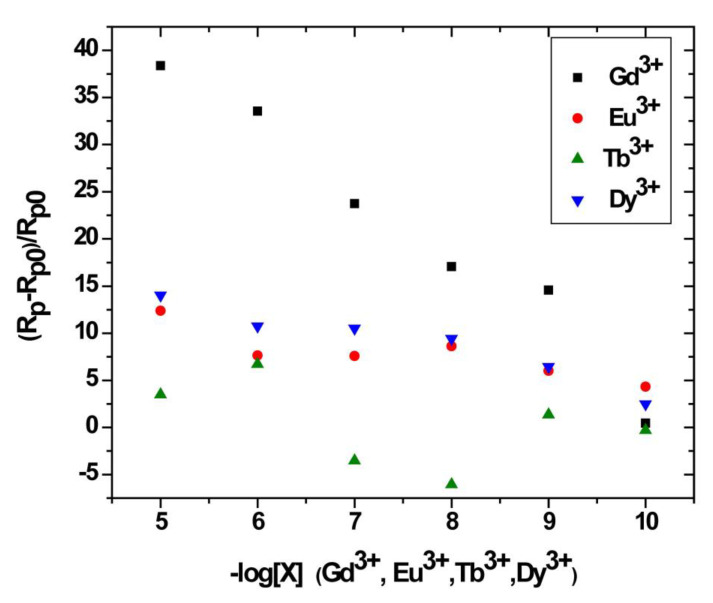
Relative variation of the polarization resistance of bis-N-MPyC sensor as a function of −log[X] for X = Gd^3+^, Eu^3+^, Tb^3+^ and Dy^3+^.

**Table 1 sensors-21-01658-t001:** Contact angle measurements of gold bare electrode, bis-N-MPyC film modified gold electrodes.

Surface	*θ*(°)			Surface Energies (mJ·m^−2^)				
	Water	Formamide	Diiodomethane	γ^s^	γ^d^	γ^p^	γ^+^	γ^−^
Gold	55.0	33.0	18.0	52.0	47.4	4.6	0.3	17.8
Gold/bis-N-MPyC	7.8	16.7	15.7	52.8	48.0	4.8	0.1	58.2

**Table 2 sensors-21-01658-t002:** Parameters of the electrical circuit equivalent to the Au/bis-N-MPyC electrode for different concentrations of lanthanides Gd^3+^, Eu^3+^, Tb^3+^ and Dy^3+^. The active area was 0.19 cm^2^.

X	[X] (mol L^−1^)	0	10^−10^	10^−9^	10^−8^	10^−7^	10^−6^	10^−5^
Gd^3+^	*R_f_* (kΩ·cm^−2^)	25.59	28.21	31.07	33.50	35.60	36.80	39.20
	*CPE_f_* (μF)	28.33	26.15	26.49	22.42	21.44	20.89	19.87
	*n_f_*	0.85	0.85	0.86	0.86	0.86	0.87	0.87
	*R_p_* (kΩ·cm^−2^)	19.62	19.71	22.20	22.97	24.28	26.20	27.15
	*CPE_dl_* (μF)	8.27	7.76	7.44	7.38	7.16	7.04	6.92
	*n_dl_*	0.98	0.98	0.98	0.99	0.99	0.98	0.98
Eu^3+^	*R_f_* (kΩ·cm^−2^)	18.25	19.74	21.31	23.41	24.95	26.60	29.61
	*CPE_f_* (μF)	38.61	35.17	32.50	31.22	29.25	27.68	25.99
	*n_f_*	0.86	0.88	0.88	0.88	0.89	0.88	0.88
	*R_p_* (kΩ·cm^−2^)	16.49	17.20	17.48	17.91	17.74	17.75	18.53
	*CPE_dl_* (μF)	10.66	10.83	10.97	10.81	11.15	10.84	10.64
	*n_dl_*	0.99	0.97	0.98	0.98	0.99	0.99	0.99
Tb^3+^	*R_f_* (kΩ·cm^−2^)	22.32	22.63	24.16	27.65	30.06	28.84	29.08
	*CPE_f_* (μF)	29.84	28.80	28.39	23.24	23.29	23.71	23.12
	*n_f_*	0.86	0.86	0.86	0.88	0.87	0.88	0.88
	*R_p_* (kΩ·cm^−2^)	18.65	18.59	18.90	17.52	17.99	19.90	19.30
	*CPE_dl_* (μF)	9.56	10.10	9.91	10.40	9.95	9.82	9.73
	*n_dl_*	0.99	0.99	0.99	0.99	0.99	0.99	0.99
Dy^3+^	*R_f_* (kΩ·cm^−2^)	19.80	20.92	21.77	24.05	26.31	25.55	26.60
	*CPE_f_* (μF)	29.18	27.48	27.87	27.01	26.00	24.47	24.03
	*n_f_*	0.87	0.87	0.87	0.87	0.87	0.88	0.89
	*R_p_* (kΩ·cm^−2^)	16.45	16.86	17.51	18.00	18.18	18.22	18.76
	*CPE_dl_* (μF)	11.98	11.94	11.51	11.21	10.85	11.39	11.15
	*n_dl_*	0.99	0.99	0.99	0.99	0.99	0.99	0.99

**Table 3 sensors-21-01658-t003:** Responses of sensors based on bis-N-MPyC films for Gd^3+^, Eu^3+^, Tb^3+^ and Dy^3+^ lanthanides and linear ranges.

Lanthanide	Linearity Range (mol L^−1^)	∆*R_p_*/*R_p_*_0_ (|S|/decade) × 100
Gd^3+^	10^−10^–10^−5^	7.3
Eu^3+^	10^−10^–10^−5^	≤1
Dy^3+^	10^−10^–10^−5^	≤1
Tb^3+^	10^−10^–10^−5^	≤1

**Table 4 sensors-21-01658-t004:** Comparison of the analytical performance of the bis-N-MPyC functionalized gold electrode with that of the previously published electrochemical sensors for gadolinium.

Electrochemical Technique	Recognition Part	Dynamic Range	Detection Limit	Reference
DP Voltammetry	MIP based on vinylpyridine	60 nM–48 µM	4.5 nM	[33]
Impedancemetry	Terpyridine ligands	10 nM–1 mM	3.5 nM	[34]
Optode	Bis(thiophenal) pyridine-2,6-diamine	50 nM–2.5 µM	9.3 nM	[46]
Optode	(Z)-N′-((Pyridine-2-yl) methylene) thiophene-2-carbohydrazide	50 nM–20 µM	10 nM	[47]
ICP-MS		5 ng/L–100 ng/L	3 ng/L; 19 pM	[48]
Impedancemetry	Methylpyridine cyclam	0.1 nM–10 µM	35 pM	This work

## Data Availability

Data are avalaible on demand to the corresponding author.

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
