# Peer review of "Detection of Gadolinium with an Impedimetric Platform Based on Gold Electrodes Functionalized by 2-Methylpyridine-Substituted Cyclam"

_sensors, 2021, doi:10.3390/s21051658_

Round 1
Reviewer 1 Report
This manuscript described an impedimetric sensor for detection lanthanides based on 2-methyl-pyridine substituted cyclam (bis-N-MPyC) modified gold electrodes. The detection mechanism was discussed and the results were supported by some data. However, a major revision is necessary for the publication.
- Why did the authors use a non-Faradic impedimetric technique in this sensor? Is there any different result with an electric probe in the underlying solution?
- Line, 218-225, why -0.9 V was chosen as the optimal voltage? Any quantitative evidence? Why 30 min was used for adsorption equilibration? The authors should provide the optimal curve.
- Figure 3, the IR-ATR data after the films complexed with lanthanide ions should be supplied to support the interactions.
- According to Figure 8, proton might play an important role in the complexing of lanthanides with the films. The dependence of EIS signals on the pH of underlying solution should be provided.
- The experimental data and fitting data should be distinguished in all Nyquist curves.
- The English grammar should be corrected in the manuscript, including superscript, tenses, e.t.
Reviewer 2 Report
This manuscript describes Detection of gadolinium with an impedimetric
platform based on gold electrodes functionalized by 2-methyl-pyridine substituted cyclam . The authors have carried out extensive characterisation studies. These results are therefore likely to be of interest to the sensing community. However, at present the work is not well-presented, and the manuscript is not up to the standard expected for publication, so significant changes are required before publication would be appropriate.
- The abstract should begin with a sentence or two about the importance of the current work.
- The authors should provide some rationale for their sensor design.
-
The superiorities of the sensor reported here also need to be demonstrated comparing to the other reported ones for the detection of gadolinium.
- How to avoid the possible interferences from the pH in different samples need to be discussed in the manuscript.
Author Response
Manuscript ID: sensors-1065432
Type of manuscript: Article
Title: Detection of gadolinium with an impedimetric platform based on gold electrodes functionalized by 2-methyl-pyridine substituted cyclam
Authors: Hassen Touzi, Yves Chevalier, Hafedh Ben Ouada, Nicole Jaffrezic-Renault *
Responses to reviewer 2’ comments
The authors thank the reviewers for the valuable comments that will improve the quality of the manuscript.
Reviewer #2
This manuscript describes Detection of gadolinium with an impedimetric
platform based on gold electrodes functionalized by 2-methyl-pyridine substituted cyclam . The authors have carried out extensive characterisation studies. These results are therefore likely to be of interest to the sensing community. However, at present the work is not well-presented, and the manuscript is not up to the standard expected for publication, so significant changes are required before publication would be appropriate.
- The abstract should begin with a sentence or two about the importance of the current work.
The following sentence was added:
Gadolinium is extensively used in pharmaceuticals and is very toxic, then it sensitive detection is mandatory.
- The authors should provide some rationale for their sensor design.
The sensor design is described in the introduction as follows:
In this work, 2-methyl-pyridine substituted cyclam (bis-N-MPyC) as a tetraazamacrocycle, was used as recognition molecule for the detection of gadolinium. It was synthesized using the bisaminal process and then immobilized as a film on gold electrodes. When a gadolinium solution is equilibrated with the modified gold electrode, due to the gadolinium complexation, the impedance of the modified gold electrode/electrolyte interface is modified, leading to a sensitive detection of gadolinium.
- The superiorities of the sensor reported here also need to be demonstrated comparing to the other reported ones for the detection of gadolinium.
The analytical performance of the sensor was compared to previously published electrochemical sensors in Table 4 and discussed in lines 481-483.
When compared to the previously published electrochemical and optical sensors for gadolinium, the detection limit of this sensor is much lower (Table 4). The detection limit is in the range of pM, as that of the sophisticated analytical technique ICP-MS.
- How to avoid the possible interferences from the pH in different samples need to be discussed in the manuscript.
The presented analytical platform is studied in 0.1 M ammonium acetate solution at pH 6.8. When the value of pH changes, the analytical performance of the platform will be modified, even the sensitivity and the selectivity. Further studies will be undertaken on this point.

Reviewer 3 Report
In regard to the manuscript ”Detection of gadolinium with an impedimetric platform based on gold electrodes functionalized by 2-methyl-pyridine substituted cyclam”
I have the following
questions:
- It was mentioned in Lines 129 and 278 in the submitted text, that cyclic voltammetry was used in the characterization of the sensors. However, one can’t find any results from cyclic voltammetry measurements!?
- In Table 2 why the resistance of the film (Rf), before contact with lanthanides (when X=0), is different for different case? Why its value is 25.59, 18.25, 22.32, and 19.8 in the case of Gd, Eu, Tb, and Dy, respectively?
- The film was deposited using the spin-coating technique. How repeatable this method was for depositing the film?
- What could be the reason that the sensor is more sensitive to Gd cation compared to the cations of other lanthanides studied in the work?
- Some of the authors of this manuscript have a published article entitled “Gold electrodes functionalized by methyl-naphthyl substituted cyclam films for the detection of metal ions”. Why this article was not included as a reference when discussing the substituted cyclams?
Comments, suggestions, and corrections:
- In some parts of the text, different fonts are used. Please check!
- Line 69: I think “the molecules” should be “ionophores” or “ionophore molecules”.
- Line 85: the sentence “Its concentration in healthy subjects e as…” should be re-written and corrected.
- Line 172: “rinsed” and “dried” should be “rinsing” and “drying”.
- Line 180: “used” should be “being used”.
- Line 183: I think “deposed” should be “deposited”?
- Line 231: the sentence should start with “Because” or “As” to make it more clear.
- Line 242: does “a whole day” mean the 8 hours mentioned in Figure 1?
- Line 283: should be “surface enhanced Raman spectroscopy (SERS)”.
- Line 562: the title of Ref [28] is missing.
Author Response
Manuscript ID: sensors-1065432
Type of manuscript: Article
Title: Detection of gadolinium with an impedimetric platform based on gold electrodes functionalized by 2-methyl-pyridine substituted cyclam
Authors: Hassen Touzi, Yves Chevalier, Hafedh Ben Ouada, Nicole Jaffrezic-Renault *
Responses to reviewer 3’ comments
The authors thank the reviewer for the valuable comments that will improve the quality of the manuscript.
Reviewer #3
In regard to the manuscript ”Detection of gadolinium with an impedimetric platform based on gold electrodes functionalized by 2-methyl-pyridine substituted cyclam”
I have the following
questions:
- It was mentioned in Lines 129 and 278 in the submitted text, that cyclic voltammetry was used in the characterization of the sensors. However, one can’t find any results from cyclic voltammetry measurements!?
These mentions were cancelled.
- In Table 2 why the resistance of the film (Rf), before contact with lanthanides (when X=0), is different for different case? Why its value is 25.59, 18.25, 22.32, and 19.8 in the case of Gd, Eu, Tb, and Dy, respectively? The film was deposited using the spin-coating technique. How repeatable this method was for depositing the film?
These values of the resistance of the film (Rf) before contact with lanthanides are extracted from the fitting data of the experimental results presented in the Nyquist plots.The mean value is 24.5± 4.1 kW·cm-2 (RSD 20%). From Ref P.L.G. Jardim et al, Applied Optics 2014, 53 (9) 1820-1825 [41], it comes that the reproducibility of the film thickness deposited by the spin-coating technique is in the range of 10%.
The discrepancy between the RSD of experimental thickness and the RSD of evaluated resistance is due to the approximation in the resistive and capacitive properties of the film. The obtained value of nf is 0.86±0.01, showing that the film is not purely capacitive, due to its roughness and its porosity.
This point was discussed in the manuscript in page 11, lines 405-410
- What could be the reason that the sensor is more sensitive to Gd cation compared to the cations of other lanthanides studied in the work?
This point was discussed in page 11, Lines 423-425.
The stability of the gadolinium complex, whose charge neutrality is ensured by acetate ions, should be due to the fastest water exchange rate reported in the literature for a neutral complex with only one inner sphere water molecule [42].
- Some of the authors of this manuscript have a published article entitled “Gold electrodes functionalized by methyl-naphthyl substituted cyclam films for the detection of metal ions”. Why this article was not included as a reference when discussing the substituted cyclams?
- This reference was cited as Ref 29 in the introduction (lines 93-94) and as previous work in lines 226 and 301.
Comments, suggestions, and corrections:
- In some parts of the text, different fonts are used. Please check! This point was corrected
- Line 69: I think “the molecules” should be “ionophores” or “ionophore molecules”. This point was corrected
- Line 85: the sentence “Its concentration in healthy subjects e as…” should be re-written and corrected. The sentence was re-written as: “Its concentration in healthy subjects is as low as 0.3 µg/L.”
- Line 172: “rinsed” and “dried” should be “rinsing” and “drying”. This point was corrected
- Line 180: “used” should be “being used”. This point was corrected
- Line 183: I think “deposed” should be “deposited”? This point was corrected
- Line 231: the sentence should start with “Because” or “As” to make it more clear. The sentence was modified as follows: “As cyclam (1,4,8,11-tetraazacyclotetradecane) contains four amine groups with the same reactivity with respect to electrophilic attack, protection of these groups is necessary to carry out an alkylation.”
- Line 242: does “a whole day” mean the 8 hours mentioned in Figure 1? These points were homogenized as 8 h.
- Line 283: should be “surface enhanced Raman spectroscopy (SERS)”. This point was corrected
- Line 562: the title of Ref [28] is missing. This point was corrected

Round 2
Reviewer 1 Report
This is a revised paper that I previously reviewed. The authors didn’t provide improvement in the text and some important problems are still confusing. The authors mentioned some explanations in the response but did not provide a rationale experimental results that are consistent with their other conclusions in the manuscript. I strongly suggest they resolve some problems with real data to confirm their assumptions rather than over interpreting their exiting results. It is the responsibility of the researchers to put forth plausible explanations for their results. Only after they do so should the paper be considered for publication.
1) The authors didn’t explain why the non-Faradic impedance was used in the main text. They claimed in the response that “the occurring redox equilibrium is: 2H+ + 2e- « H2”, however, the underlying pH is 6.8, near neutral. The concentration of H+ is much lower than that of the supporting electrolytes in the buffer. Thus, the equivalent circuit showed in Figure 7 is totally wrong. 2) If the H+ acts as a redox in the EIS testing, and the H+ also plays an important role in the recognition Gd3+ in the mechanism in Figure 8, the results depending on the pH of underlying solution must be provided for this sensor. 3) The authors claimed in the response that “IR-ATR spectrum was obtained after lanthanide adsorption, no modification of the IR-ATR spectrum was observed”. I don’t agree with that. As Figure 8 showed, if the coordination bond linked between Gd3+ and -N-, -O- atoms in the cyclam ligands, the stretching vibration of corresponding groups containing -N-, -O- must be changed. The resolution of IR-ATR spectrum might be not enough. Derivative or second derivative of IR data is suggested to improve the resolution.
Author Response
Please see attached file where figures are included

Reviewer 2 Report
Recommending acceptance after typos correction.
Author Response
Manuscript title: Detection of gadolinium with an impedimetric platform based on gold electrodes functionalized by 2-methylpyridine-substituted cyclam
Responses to reviewer 2’ comments
The authors thank the reviewer for the valuable comments that will improve the quality of the manuscript.
Reviewer #2
Recommending acceptance after typos correction.
Typos mistakes were corrected.
